# Cross-lingual Transfer Learning for COVID-19 Outbreak Alignment

**Sharon Levy** and **William Yang Wang**
University of California, Santa Barbara
Santa Barbara, CA 93106
{sharonlevy,william}@cs.ucsb.edu

## Abstract

The spread of COVID-19 has become a significant and troubling aspect of society in 2020. With millions of cases reported across countries, new outbreaks have occurred and followed patterns of previously affected areas. Many disease detection models do not incorporate the wealth of social media data that can be utilized for modeling and predicting its spread. It is useful to ask, can we utilize this knowledge in one country to model the outbreak in another? To answer this, we propose the task of cross-lingual transfer learning for epidemiological alignment. Utilizing both macro and micro text features, we train on Italy's early COVID-19 outbreak through Twitter and transfer to several other countries. Our experiments show strong results with up to 0.85 Spearman correlation in cross-country predictions.

## 1 Introduction

During the COVID-19 pandemic, society was brought to a standstill, affecting many aspects of our daily lives. With increased travel due to globalization, it is intuitive that countries have followed earlier affected regions in outbreaks and measures to contain to them (Cuffe and Jeavans, 2020).

A unique form of information that can be used for modeling disease propagation comes from social media. This can provide researchers with access to unfiltered data with clues as to how the pandemic evolves. Current research on the COVID-19 outbreak concerning social media includes word frequency and sentiment analysis of tweets (Rajput et al., 2020) and studies on the spread of misinformation (Kouzy et al., 2020; Singh et al., 2020). Social media has also been utilized for other disease predictions. Several papers propose models to identify tweets in which the author or nearby person has the attributed disease (Kanouchi et al., 2015; Aramaki et al., 2011; Lamb et al., 2013; Kitagawa et al., 2015). Iso et al. (2016) and Huang et al.

(2016) utilize word frequencies to align tweets to disease rates. A shortcoming of the above models is they do not consider how one region's outbreak may relate to another. Many of the proposed models also rely on lengthy keyword lists or syntactic features that may not generalize across languages. Text embeddings from models such as multilingual BERT (mBERT) (Devlin et al., 2019) and LASER (Artetxe and Schwenk, 2019) can allow us to combine features and make connections across languages for semantic alignment.

We present an analysis of Twitter usage for cross-lingual COVID-19 outbreak alignment. We study the ability to correlate social media tweets across languages and countries in a pandemic scenario. Based on this demonstration, researchers can study various cross-cultural reactions to the pandemic on social media. We aim to analyze how one country's tweets align with its own outbreak and if those same tweets can be used to predict the state of another country. This can allow us to determine how actions taken to contain the outbreak can transfer across countries with similar measures. We show that we can achieve strong results with cross-lingual transfer learning.

Our contributions include:

- We formulate the task of cross-lingual transfer learning for epidemiological outbreak alignment across countries.

- We are the first to investigate state-of-the-art cross-lingual sentence embeddings for cross-country epidemiological outbreak alignment. We propose joint macro and micro reading for multilingual prediction.

- We obtain strong correlations in domestic and cross-country predictions, providing us with evidence that social media patterns in relation to COVID-19 transcend countries.

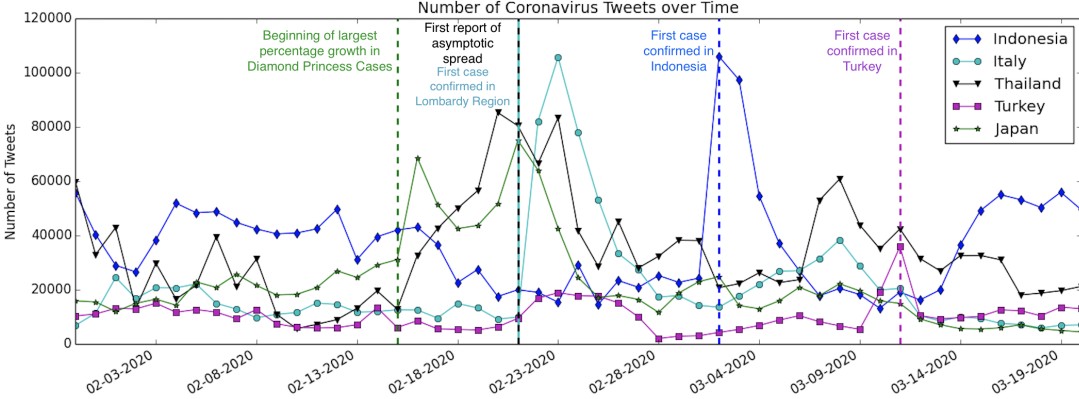

Figure 1: Timeline of COVID-19-related tweets, from COVID-19 dataset (Chen et al., 2020), in various languages. The peaks are marked by events relating to each language's main country's initial outbreak.

## 2 Twitter and COVID-19

### 2.1 Problem Formulation

An intriguing question in the scope of epidemiological research is: can atypical data such as social media help us model an outbreak? To study this, we utilize Twitter as our source, since users primarily post textual data and in real-time. Furthermore, Twitter users transcend several countries, which is beneficial as COVID-19 is analyzed by researchers and policymakers on a country by country basis (Kaplan et al., 2020). Our motivation in this paper is the intuition that social media users can provide us with indicators of an outbreak during the COVID-19 pandemic. In this case, we reformulate our original question: can we align Twitter with a country's COVID-19 outbreak and apply the learned information to other countries?

### 2.2 Data

We utilize the COVID-19 Twitter dataset (Chen et al., 2020), comprised of millions of tweets in several languages. These were collected through Twitter's streaming API and Tweepy[1] by filtering for 22 specific keywords and hashtags related to COVID-19 such as Coronavirus, Wuhanlockdown, stayathome, and Pandemic. We consider tweets starting from February 1st, 2020 to April 30th, 2020, and filter for tweets written in Italian, Indonesian, Turkish, Japanese, and Thai. Specifically, we filter for languages that are primarily spoken in only one country, as opposed to languages such as English and Spanish that are spoken in several countries. In Table 1, we show dataset statistics describing total tweet counts for each country along

[1] https://www.tweepy.org/

|  | Italy | Thailand | Japan | Turkey | Indonesia |
|---|---|---|---|---|---|
| Pre | 1.3M | 2.2M | 2.2M | 960K | 3.2M |
| Post | 103K | 6.9K | 61K | 96K | 309K |

Table 1: Dataset statistics in each country before (Pre) and after (Post) the tweet filter process described in Section 2.5.

with counts after our filtering process described later in Section 2.5. When aligning tweets with each country's outbreak, we utilize the COVID-19 Dashboard by the CSSE at Johns Hopkins University (Dong et al., 2020) for daily confirmed cases from each country. Since the COVID-19 pandemic is still in its early stages at the time of writing this paper, sample sizes are limited. Therefore, our experiments have the following time cut settings: train in February and March and test in April (I), train in February and test in March and April (II), train in February and test in March (III), and train in March and test in April (IV).

### 2.3 Can Twitter detect the start of a country's outbreak?

We start by investigating a basic feature in our dataset: tweet frequency. We plot each country's tweet frequency in Figure 1. There is a distinct peak within each country, corresponding to events within each country signaling initial outbreaks, denoted by the vertical lines. These correlations indicate that even a standard characteristic such as tweet frequency can align with each country's outbreak and occurs across several countries. Given this result, we further explore other tweet features for epidemiological alignment.

## 2.4 Cross-Lingual Transfer Learning

We determine that it is most helpful for researchers to first study regions with earlier outbreaks to make assumptions on later occurrences in other locations. In this case, Italy has the earliest peak in cases. When aligning outbreaks from two different countries, we experiment with the transfer learning setting. We train on Italy's data and test on the remaining countries. We attempt to answer whether we can build a model that correlates the day's tweets with the number of cases in a given country and if we can apply this trained model to tweets and cases in a new country with a different language and culture.

We present this as a regression problem in which we map our input text features $\mathbf{x} \in \mathbb{R}^n$ to the output $\mathbf{y} \in \mathbb{R}$. Our ground-truth output $\mathbf{y}$ is presented in two scenarios in our experiments: total cases and daily new cases. The former considers all past and current reported cases while the latter consists of only cases reported on a specific day. The predicted output $\hat{\mathbf{y}}$ is compared against ground truth $\mathbf{y}$. During training and test time, we utilize support vector regression for our model and concatenate the chosen features as input each day. Due to different testing resources, criteria, and procedures, there are some offsets in each countries' official numbers. Therefore, we follow related disease prediction work and evaluate predictions with Spearman's correlation (Hogg et al., 2005) to align our features with official reported cases.

## 2.5 Creating a Base Model

In the wake of the COVID-19 crisis, society has adopted a new vocabulary to discuss the pandemic (Katella, 2020). Quarantine and lockdown have become standard words in our daily conversations. Therefore, we ask: are there specific features that indicate the state of an outbreak?

**Which features can we utilize for alignment?** We create a small COVID-19-related keyword list consisting of lockdown, quarantine, social distancing, epidemic, and outbreak and translate these words into Italian. We include the English word "lockdown" as it has been used in other countries' vocabularies. We aim to observe which, if any, of these words align with Italy's outbreak. In addition to word frequencies, we also utilize mBERT and LASER to extract tweet representations for semantic alignment. We remove duplicate tweets, retweets, tweets with hyperlinks, and tweets dis-

| Cases | Embed | Time Setting | | | |
|---|---|---|---|---|---|
| | | I | II | III | IV |
| Total | mBERT | **0.880** | **0.947** | **0.769** | **0.880** |
| | LASER | 0.879 | 0.946 | 0.766 | 0.879 |
| New | mBERT | **0.805** | 0.416 | 0.718 | 0.794 |
| | LASER | 0.800 | **0.490** | **0.723** | **0.800** |

Table 2: Italy's Spearman correlation results with total and daily case count prediction for mBERT and LASER (Embed). Time settings are defined in 2.2. We bold the highest correlations within each case setting.

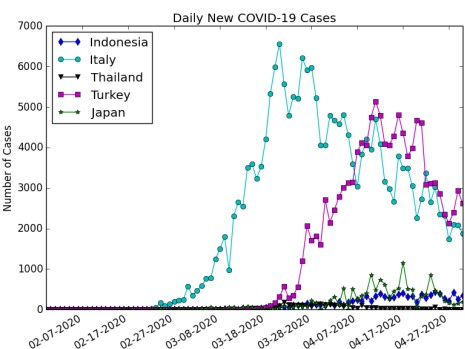

Figure 2: Distribution of new daily COVID-19 cases in Italy, Turkey, Thailand, Japan, and Indonesia. Daily case counts come from COVID-19 Dashboard by CSSE at Johns Hopkins University (Dong et al., 2020).

cussing countries other than Italy (tweets with other country names) in order to focus more on personal narratives within the country. Using the sentence encoding service bert-as-a-service (Xiao, 2018), we extract fixed-length representations for each tweet. We explore two options for our tweet representations: average-pooling and max-pooling. Our final feature consists of daily tweet frequency after filtering.

**Can tweet text align with confirmed cases?** We combine combinations of our frequency features with our tweet embeddings and show results in Table 2. Through manual tuning, we find our strongest model (polynomial kernel) contained the English keyword lockdown and averaged tweet representations from mBERT for the total case scenario. When aligning to new cases, the best model (sigmoid kernel) contained the English keyword lockdown and max-pooled LASER embeddings. While mBERT and LASER provide very little difference in alignment to total cases, LASER is noticeably stronger in the new case setting, particularly in II. For the total case setting, our predictions show strong alignment with ground truth, which is monotonically increasing, in all time settings. When measuring new daily cases, the correlations

| Setting | Thailand | Japan | Turkey | Indonesia |
|---------|----------|-------|--------|-----------|
| I   | 0.200 | -.300 | .188  | -.316 |
| II  | 0.696 | 0.543 | 0.715 | 0.285 |
| III | 0.823 | 0.856 | 0.679 | 0.925 |
| IV  | 0.196 | -.300 | 0.188 | -.316 |
| V   | 0.859 | 0.649 | 0.817 | 0.722 |

Table 3: Cross-lingual transfer learning Spearman correlation with total case counts while training with Italy data. Time settings are defined in 2.2.

| Setting | Thailand | Japan | Turkey | Indonesia |
|---------|----------|-------|--------|-----------|
| I   | -.022 | 0.130 | -.368 | 0.416 |
| II  | 0.277 | 0.273 | 0.426 | 0.332 |
| III | 0.661 | 0.262 | 0.255 | 0.407 |
| IV  | -.043 | 0.127 | -.375 | 0.416 |
| V   | 0.755 | 0.515 | 0.745 | 0.742 |

Table 4: Cross-lingual transfer learning Spearman correlation with new daily case counts while training with Italy data. Time settings are defined in 2.2.

are weaker in II. We find that Italy's new cases form a peak in late March, as shown in Figure 2. As a result, there is a distribution shift when training on February data only (tail of the distribution) and testing in March and April.

## 2.6 Cross-Lingual Prediction

While we can align historical data to future cases within Italy, researchers may not have enough data to train models for each country. Therefore we ask, can we use Italy's outbreak to predict the outbreak of another country? In particular, we determine whether users from two different countries follow similar patterns of tweeting during their respective pandemics and how well we can align the two. We follow the same tweet preprocessing methodology described in Section 2.5 and the timeline cuts for training and testing defined in Section 2.2. We also add another time setting (V): training in February, March, and April and testing all three months. This serves as an upper bound for our correlations, indicating how well the general feature trends align between the two countries and their outbreaks.

**Can we transfer knowledge to other countries?** We show our results for the total and new daily case settings in Tables 3 and 4. All of the test countries have strong correlations in time setting V for both case settings. Since this is used as an upper bound, we can deduce that tweets across countries follow the same general trend in relation to reported cases. When examining the other time settings, it is clear that Italy transfers well in II and III for the total

case setting. As these train in February only, this shows us that transferring knowledge works better in times of more linear case increases, rather than during peaks, which becomes unstable. Times I through IV generally do not perform as well in the new case setting, though II and III primarily have higher correlations.

**Why does Indonesia differ?** It is noticeable that Indonesia aligns better with new daily cases in times I through IV, as opposed to the other countries. When examining Figure 2, we find that Indonesia is the only country that had not yet reached a peak in new daily cases by the end of April, and is steadily increasing. Meanwhile, the other countries follow normal distributions like Italy. However, given that we train our model on February and March data, it does not learn information on post-peak trends and cannot generalize well to these scenarios that occur in April in the other countries.

**What can we learn from our results?** Overall, transfer learning in the total case setting leads to stronger correlations with case counts. While results show that training in February and testing in March and/or April works best, our results for V's upper bound correlation show that weaker correlations can be due to the limited sample sizes we have from the start of the pandemic. Additionally, training in February, March, and April in Italy allows us to model a larger variety of scenarios during the pandemic, with samples during pre, mid, and post-peak. Therefore, as we obtain more data every day, we can build stronger models that can generalize better to varying distributions of cases and align outbreaks across countries that can fully reach their upper bound correlations and beyond.

## 3 Conclusion

In this paper, we performed an analysis of cross-lingual transfer learning with Twitter data for COVID-19 outbreak alignment using cross-lingual sentence embeddings and keyword frequencies. We showed that even with our limited sample sizes, we can utilize knowledge of countries with earlier outbreaks to correlate with cases in other countries. With larger sample sizes and when training on a variety of points during the outbreak, we can obtain stronger correlations to other countries. We hope our analysis can lead to future integration of social media in epidemiological prediction across countries, enhancing outbreak detection systems.

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
