# OpenReview forum: "Cross-lingual Transfer Learning for COVID-19 Outbreak Alignment"
_aclweb.org/ACL/2020/Workshop/NLP-COVID — NLP-COVID-2020 Abstractonly_

### Official Review · AnonReviewer1 · 2020-06-11
**Review of: Cross-lingual Transfer Learning for COVID-19 Outbreak Alignment**

**Rating:** 7
**Confidence:** 3

**Review:**

# [REVIEW] Cross-lingual Transfer Learning for COVID-19 Outbreak Alignment

10th June 2020

## SUMMARY
This short paper describes work on cross-lingual transfer learning to track COVID-19 cases across Italy, Thailand, Japan, Turkey, and Indonesia (i.e. languages that are largely spoken in a single country).  Using case statistics derived from Johns Hopkins COVID dashboard, the researchers aim to determine (a) if the number of COVID-related tweets is associated with case numbers (good correlations were achieved for this), and (b) can this training be applied to other countries that may be at a slightly different point in their COVID trajectory i.e. cross-lingual transfer learning (a range of correlations with JHU data were achieved ranging from -.316 to 0.859, indicating that the approach has some utility — but see caveats below).

It is difficulty for me to assess the quality of this paper relative to other submissions, but I suspect that this use of cross-lingual transfer learning is sufficiently interesting to justify a short paper.

## MAJOR COMMENTS
1.  There is a major methodological issue not related to the NLP and not something you can be reasonably expected to do anything about — but I think it’s important to point it out as a limitation:  The international ground truth data for COVID is  not very reliable regarding underlying prevalence/incidence of COVID-19.  Even in the US, there have been stark differences in approaches to testing, and the criteria for testing. International comparisons are even more difficult. This undermines the evaluation somewhat.

## MINOR COMMENTS [NITPICKING]
1.  [p1c1] “With globalization, it is intuitive that countries have followed earlier affected regions in patterns of outbreaks and measures to contain to them (Cuffe and Jeavans, 2020).” Consider rephrasing this to emphasise that it is the increased travel associated with globalization that is important, rather than globalisation per se.
2.  [p3c1] “Bert as service” probably requires a few words of  explanation (at least for this reviewer - I had to google it)
3.  [p3c2] “When measuring against new daily cases, the correlations are not as significant in time II”  suggest avoiding using the word significance here, unless you are referring to statistical significance.

---

> ### Author Response · Authors · 2020-06-24
> **Reply to Reviewer 1**
>
> Thank you for reviewing our paper. We hope our comments below address your concerns.
>
> >There is a major methodological issue not related to the NLP and not something you can be reasonably expected to do anything about — but I think it’s important to point it out as a limitation:  The international ground truth data for COVID is  not very reliable regarding underlying prevalence/incidence of COVID-19.  Even in the US, there have been stark differences in approaches to testing, and the criteria for testing. International comparisons are even more difficult. This undermines the evaluation somewhat.
>
> We understand that due to different testing resources, criteria, and procedures, there are some offsets in each countries' official numbers. That is why we are using Spearman correlation instead of mean absolute error for the evaluation. We have tested the cross-lingual model on four countries to show that we can transfer knowledge even in the case of variable testing practices.
>
> >1.  [p1c1] “With globalization, it is intuitive that countries have followed earlier affected regions in patterns of outbreaks and measures to contain to them (Cuffe and Jeavans, 2020).” Consider rephrasing this to emphasise that it is the increased travel associated with globalization that is important, rather than globalisation per see.
> 2.  [p3c1] “Bert as service” probably requires a few words of  explanation (at least for this reviewer - I had to google it)
> 3.  [p3c2] “When measuring against new daily cases, the correlations are not as significant in time II”  suggest avoiding using the word significance here, unless you are referring to statistical significance.
>
> We have rewritten these sentences in our revision.

---

### Official Review · AnonReviewer2 · 2020-06-18
**Good idea but some methodological concerns**

**Rating:** 5
**Confidence:** 3

**Review:**

The following paper hypothesizes that tweets can be used to model COVID-19 outbreaks in across countries. The paper does this by using cross-lingual sentence embeddings from mBERT and LASER to predict case-counts.

Although this is a good idea and Figure 1 is particularly compelling, I find that the paper is at best a work in progress and at worst intentionally misleading. I'd recommend substantially more work before it is ready for publication.

My biggest concern is that the way the tweets are filtered and the time-chunks are specified, I have no reason to believe that this model is using tweets to actually predict an outbreak, as the authors imply (NB the authors are careful to say they are "aligning", not "predicting", although I think this intentionality is rather muddy). It seems to me that they are simply capturing the response to an outbreak already occurring -- I'm unclear what the value of this is to social scientists and policymakers.

The authors explicitly filter for words such as "lockdown", "quarantine", "social distancing", "epidemic", and "outbreak", to perform their alignment, which to me seem to describe tweet-responses to policy rather than tweet-responses to sickness. I think the authors' hypothesis would be better served by choosing words, topics or other indicators that are more personal and health-related -- words like "fever", "cough" or "lack of smell" -- to avoid such confounding.

Further, the authors utilized fixed time-periods to explore their regression, which is a little confusing to me. I think the authors should use date-ranges in each country relative to the kth case in that country. Since their date-ranges are so wide, I worry that the date-ranges are simply capturing the full policy response of a government that has already predicted an outbreak.

Why is spearman's correlation the only metric used? If the authors are conducting a regression experiment, there are other more compelling and interpretable metrics. Why aren't significance-values included for the correlation?

Why do the authors say that they observe "right-skewed Gaussians"? This is clearly more of a point process (i.e. a Hawkes process.)

Additionally, I would urge a more serious consideration of other confounders as well. I'm not sure what the best ones to use would be, but I think some policy analysis is necessary -- when did each country actually institute a lockdown? Do lockdowns effect overall non-COVID tweet-volume as well? I'm sure there are confounders in the literature.

If this is not possible, then I urge a reframing of the paper. The authors need to be clearer about what they are actually purporting to do, and not hide behind words like "align".

---

> ### Author Response · Authors · 2020-06-24
> **Reply to Reviewer 2**
>
> Thank you for the review. We provide a response to these comments below and hope this response and our revision sufficiently address your concerns.
>
> >My biggest concern is that the way the tweets are filtered and the time-chunks are specified, I have no reason to believe that this model is using tweets to actually predict an outbreak, as the authors imply (NB the authors are careful to say they are "aligning", not "predicting", although I think this intentionality is rather muddy). It seems to me that they are simply capturing the response to an outbreak already occurring -- I'm unclear what the value of this is to social scientists and policymakers.
>
> The model in the paper describes real-time alignment of tweets to confirmed cases in each country. The use of cross-lingual transfer learning can allow us to determine how actions taken to contain the outbreak, such as lockdowns, can transfer across countries who implement the same measures through social media sensing. Studying a country with an early outbreak can help us to detect similar COVID-19 trends in other countries.  Additionally, it allows us to study the various cultural reactions to the pandemic on social media.
>
> >The authors explicitly filter for words such as "lockdown", "quarantine", "social distancing", "epidemic", and "outbreak", to perform their alignment, which to me seem to describe tweet-responses to policy rather than tweet-responses to sickness. I think the authors' hypothesis would be better served by choosing words, topics or other indicators that are more personal and health-related -- words like "fever", "cough" or "lack of smell" -- to avoid such confounding.
>
> While we have taken the approach of analyzing policy actions in tweets for this model, our model can also be changed and utilized for the approach of symptom detection.
>
> >Further, the authors utilized fixed time-periods to explore their regression, which is a little confusing to me. I think the authors should use date-ranges in each country relative to the kth case in that country. Since their date-ranges are so wide, I worry that the date-ranges are simply capturing the full policy response of a government that has already predicted an outbreak.
>
> Our model prediction utilizes only the tweet features and does not rely on past case counts/time periods for alignment. Therefore, we can freely use any date-range instead of partitioning based on the kth case count which could potentially lead to overfitting.
>
> >Why is spearman's correlation the only metric used? If the authors are conducting a regression experiment, there are other more compelling and interpretable metrics. Why aren't significance-values included for the correlation?
>
> We emphasize that the goal of this current study is to determine if there is an alignment between tweets and confirmed cases. Due to different testing resources, criteria, and procedures, there are some offsets in each countries' official numbers. That is why we are using Spearman correlation instead of mean absolute error for the evaluation.
>
> >Why do the authors say that they observe "right-skewed Gaussians"? This is clearly more of a point process (i.e. a Hawkes process.)
>
> We use this to describe the shape of confirmed cases. However, a Hawkes process may be used to describe the diffusion of cases as seen in the figure.
>
> >Additionally, I would urge more serious consideration of other confounders as well. I'm not sure what the best ones to use would be, but I think some policy analysis is necessary -- when did each country actually institute a lockdown? Do lockdowns effect overall non-COVID tweet-volume as well? I'm sure there are confounders in the literature.
>
> While we focus only on textual features in this model, future work can include other features such as restriction measures.

---

### Official Review · AnonReviewer5 · 2020-07-06
**Interesting correlation, but not actionable and not clear**

**Rating:** 5
**Confidence:** 4

**Review:**

This paper analyzes the correlation between Covid19 cases (as reported by JHU) and relevant tweets, on a per-country based granularity.
The results are remarkable, showing a very strong correlation (Table 2) for a model trained on Italian data. Moreover, using modern multi-lingual embeddings, the trained regression carries over to other languages; although the correlation there is very variable (and sometimes negative).
The results are intriguing and show once more how expressions on social media reflect physical reality.

I have however two major concerns:

1/ Why is this useful?
Not in the sense of what could public health officials do with such models (although I have no clue what they could do). It seems to me (and please correct me if I am wrong) that you are using the tweets of day D to predict number of cases of day D. If that is the case, then the causality is the other way round (cases => tweets). It would be much more interesting to try to predict number of cases at day D+1, or even better at day D+n.

2/ Reproducibility
There are lots of things that are not clear. Those are not fundamental issues, but I cannot recommend acceptance in the current state, as I could not understand exactly what the authors did, much less would I be able to reproduce it
Filtering:
  - why do you remove tweets with hyperlinks?
 - "we remove [...] tweets discussing countries other than Italy". What does this mean?
 - "We further filter Italy’s tweets for a balanced representation of tweet embeddings." What does this mean?
Model:  what exactly is a model? Is it the embedding + the total frequency of the selected words? Could you enumerate all the "models" (feature set) you used and - maybe in the appendix - the values for all of them?
Evaluation: Why do you not report mean error? The correlation itself is not very actionable, the predicted value would be

Also, there is a confusion between language and country, which is conflated. The given argument is that the chosen languages have a majority country, but for rigourousity I would recommend replacing country everywhere by language as it is misleading.

Fig 1 seems to be done with the number of tweets before the filtering of Sect 2.5. Why?

Finally, I am not sure if I understand how the transfer is done. Is it correct to say that you use the same regression and provide as feature the Japanese/Indonesian/etc tweets after putting them through the multi-lingual embedding?

---

> ### Author Response · Authors · 2020-07-07
> **Reply to Reviewer 5**
>
> Thank you for the review. We hope our comments sufficiently address your concerns.
>
> >Not in the sense of what could public health officials do with such models (although I have no clue what they could do). It seems to me (and please correct me if I am wrong) that you are using the tweets of day D to predict number of cases of day D. If that is the case, then the causality is the other way round (cases => tweets). It would be much more interesting to try to predict number of cases at day D+1, or even better at day D+n.
>
> The model in the paper describes real-time alignment of tweets to confirmed cases in each country. Our model prediction utilizes only the tweet features and does not rely on past case counts/time periods for alignment. We study the ability to correlate social media tweets across languages and countries in a pandemic scenario. Based on this demonstration, researchers can study various cross-cultural reactions to the pandemic on social media. The use of cross-lingual transfer learning can allow us to determine how actions taken to contain the outbreak, such as lockdowns, can transfer across countries who implement the same measures through social media sensing. Studying a country with an early outbreak can help us to detect similar COVID-19 trends in other countries.
>
> >why do you remove tweets with hyperlinks?
>
> This is done to remove tweets spreading news information and focus more on personal tweets.
>
> >"we remove [...] tweets discussing countries other than Italy". What does this mean?
>
> For the Italy model, we are evaluating against reported cases in Italy. Therefore, we do not want to utilize tweets that focus on other countries as this will not have effect on Italy’s COVID-19 case status and our model focuses on the personal narratives of those living in Italy. Similarly, for other countries, we remove tweets that discuss countries other than the one being tested on.
>
> >"We further filter Italy’s tweets for a balanced representation of tweet embeddings." What does this mean?
>
> There may be duplicate tweets present in our initial data. We collapse these to a single tweet.
>
> >Model:  what exactly is a model? Is it the embedding + the total frequency of the selected words? Could you enumerate all the "models" (feature set) you used and - maybe in the appendix - the values for all of them?
>
> The model represents the trained SVM model for Italy. The input features are the embedding + the total frequency of the selected words. We have two regression models in this case: one for the number of new daily cases and another for the total case count.
>
> >Evaluation: Why do you not report mean error? The correlation itself is not very actionable, the predicted value would be
>
> Due to different testing resources, criteria, and procedures, there are some offsets in each countries' official numbers. That is why we are using Spearman correlation instead of mean absolute error for the evaluation.
>
> >Fig 1 seems to be done with the number of tweets before the filtering of Sect 2.5. Why?
>
> This is done as an initial analysis to determine whether even a basic feature such as the original tweet frequency can give us insight into the pandemic case trends.
>
> >Finally, I am not sure if I understand how the transfer is done. Is it correct to say that you use the same regression and provide as feature the Japanese/Indonesian/etc tweets after putting them through the multi-lingual embedding?
>
> We are utilizing the trained Italy regression model and using Japanese/Indonesian/etc tweets (along with keyword counts) as input to the model for testing. mBERT and LASER are used to extract multi-lingual tweet embeddings.

---

> > ### Comment · AnonReviewer5 · 2020-07-07
> > **Clarification questions**
> >
> > Thank you for this very prompt answer. It clarifies most of my doubts. For the others however, I realized it was not clear in my review the information I would have liked you to provide. I will try to be more explicit
> >
> > > Our model prediction utilizes only the tweet features and does not rely on past case counts/time periods for alignment
> > Do you confirm that for day D you use tweets from that day D to predict counts of day D?
> >
> > > For the Italy model, we are evaluating against reported cases in Italy. Therefore, we do not want to utilize tweets that focus on other countries as this will not have effect on Italy’s COVID-19 case status and our model focuses on the personal narratives of those living in Italy. Similarly, for other countries, we remove tweets that discuss countries other than the one being tested on.
> > I understand what you want to achieve. I do not understand *how* you achieve this. Could you be more clear on how you "remove countries other than the one being tested on"?
> >
> > > There may be duplicate tweets present in our initial data. We collapse these to a single tweet.
> > So, do you confirm that "filter Italy’s tweets for a balanced representation of tweet embeddings" means removing duplicates?
> >
> > > Due to different testing resources, criteria, and procedures, there are some offsets in each countries' official numbers. Even inside the same country? Wouldn't that affect the correlation as well? I am not convinced of this argument, and still believe that the real value lies in the predicted value themselves. Could you report those in the appendix for instance?

---

> > > ### Author Response · Authors · 2020-07-09
> > > **Response to Clarification Questions**
> > >
> > > >Our model prediction utilizes only the tweet features and does not rely on past case counts/time periods for alignment Do you confirm that for day D you use tweets from that day D to predict counts of day D?
> > >
> > > Yes. In this paper, we first attempt to answer whether we can build a model that can correlate the day's tweets with the number of cases in a given country. We then attempt to see if we can use this trained model and apply it to tweets and cases in a new country with a different language and culture.
> > >
> > > >For the Italy model, we are evaluating against reported cases in Italy. Therefore, we do not want to utilize tweets that focus on other countries as this will not have effect on Italy’s COVID-19 case status and our model focuses on the personal narratives of those living in Italy. Similarly, for other countries, we remove tweets that discuss countries other than the one being tested on. I understand what you want to achieve. I do not understand how you achieve this. Could you be more clear on how you "remove countries other than the one being tested on"?
> > >
> > > When we do this, we simply remove tweets that include other countries’ names when translated into the current country’s language.
> > >
> > > >There may be duplicate tweets present in our initial data. We collapse these to a single tweet. So, do you confirm that "filter Italy’s tweets for a balanced representation of tweet embeddings" means removing duplicates?
> > >
> > > Yes, filtering for a balanced tweet representation is the process of removing duplicate tweets.
> > >
> > > >Due to different testing resources, criteria, and procedures, there are some offsets in each countries' official numbers. Even inside the same country? Wouldn't that affect the correlation as well? I am not convinced of this argument, and still believe that the real value lies in the predicted value themselves. Could you report those in the appendix for instance?
> > >
> > > Yes, there could be the case of different testing resources within a country and this may introduce some noise into the data.

---

> > > > ### Comment · AnonReviewer5 · 2020-07-10
> > > > **Thanks & Congrats**
> > > >
> > > > Thanks a lot for getting back, even after acceptance (I just watched your talk).
> > > > In case you continue this work, I can only encourage you to look into predictions for days D+n, as well as reporting the numbers of predicted cases in addition to correlations.

---

### Decision · Program_Chairs · 2020-07-07

**Decision:**

Accept (Abstract only)

**Comment:**

While the reviewers raise some methodological concerns, the methodology and results presented are interesting and worthy of dissemination, as well as being adequate for a short paper. It is my view that the concerns can be addressed in a revision of the paper (to be prepared for the final proceedings volume, after the workshop).

We look forward to a presentation of this work at the workshop on Thursday.